# Improving dermatology classifiers across populations using images generated by large diffusion models

**Luke W. Sagers**[*]
Department of Biomedical Informatics
Harvard Medical School
luke_sagers@hms.harvard.edu

**James A. Diao**[*]
Department of Biomedical Informatics
Harvard Medical School
james_diao@hms.harvard.edu

**Matthew Groh**
MIT Media Lab
Massachussetts Institute of Technology
groh@mit.edu

**Pranav Rajpurkar**
Department of Biomedical Informatics
Harvard Medical School
pranav_rajpurkar@hms.harvard.edu

**Adewole S. Adamson**
Department of Internal Medicine
UT Austin Dell Medical School
adewole.adamson@austin.utexas.edu

**Arjun K. Manrai**
Department of Biomedical Informatics
Harvard Medical School
arjun_manrai@hms.harvard.edu

## Abstract

Dermatological classification algorithms developed without sufficiently diverse training data may generalize poorly across populations. While intentional data collection and annotation offer the best means for improving representation, new computational approaches for generating training data may also aid in mitigating the effects of sampling bias. In this paper, we show that DALL·E 2, a large-scale text-to-image diffusion model, can produce photorealistic images of skin disease across skin types. Using the Fitzpatrick 17k dataset as a benchmark, we demonstrate that augmenting training data with DALL·E 2-generated synthetic images improves classification of skin disease overall and especially for underrepresented groups.

## 1  Introduction

Skin disease classification algorithms based on modern machine learning algorithms are now entering clinical and commercial use [4, 8, 14]. Although these algorithms have demonstrated results on par with those of board-certified dermatologists, many remain concerned about the limited representation of darker skin tones in development datasets, which may negatively impact their performance across diverse populations [1, 3].

In response, several groups have collected and released benchmark data repositories with data from more diverse populations; these include skin tone annotations that may be used to study and improve bias in classification models [2, 6]. However, even with intentional upsampling in data collection, it is difficult to achieve parity in representation or performance for underrepresented groups, particularly across the dozens or hundreds of skin conditions that may be included as possible prediction outputs.

Recent breakthroughs in large-scale diffusion models [10, 12, 13] have enabled generation of photorealistic images based on either text alone or text and image inputs in combination. Synthetic data produced by such models have the potential to improve prediction models [9], especially for

---

[*]Equal contribution

NeurIPS 2022 Workshop on Synthetic Data for Empowering ML Research.

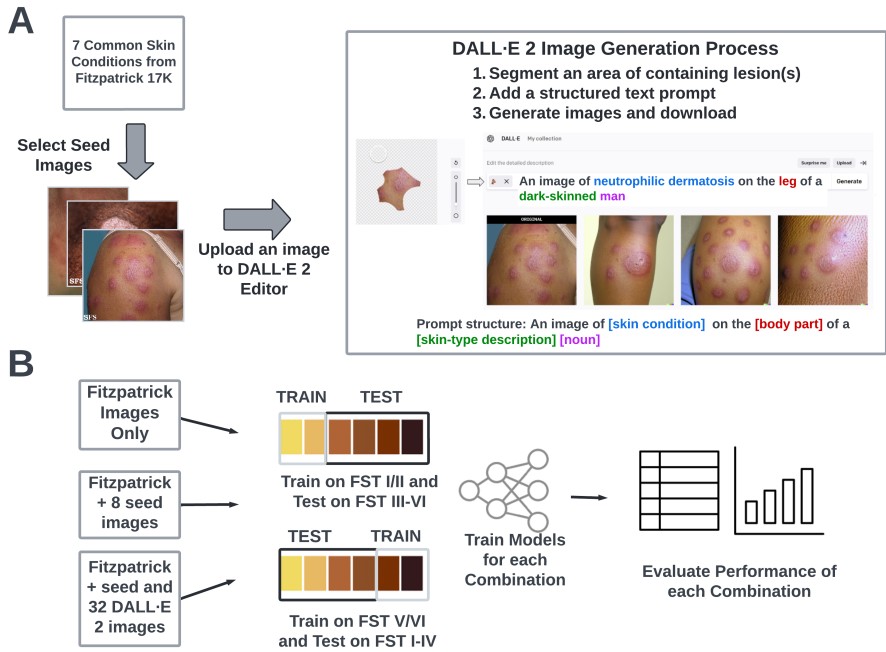

Figure 1: **A schematic overview of the study A)** We selected seven skin conditions from the Fitzpatrick 17k dataset [6]. For each skin condition, we randomly sampled eight images from the lightest and darkest Fitzpatrick skin types (16 total images for one condition) and used these images as seed images in the editor of OpenAI's DALL·E 2 model to produce synthetic variations. We selected four synthetic images per seed image for inclusion in the subsequent analysis. **B)** We trained image classification models to predict skin condition labels using different train/test splits. Models were trained on skin images from lighter skin types (FST I-II) and tested on images with darker skin types (FST V-VI), and vice versa. We ran experiments where training data included either (1) Fitzpatrick 17k images only, (2) Fitzpatrick 17k + seed images, and (3) Fitzpatrick 17k + seed images + DALL·E 2 generated synthetic images.

underrepresented disease classes and skin tones. In this paper, we describe a pipeline for producing photorealistic images of skin disease using the transformer-based generative model DALL·E 2. We show that targeted generation of synthetic images can be used to improve the performance of dermatological classifiers on a diverse benchmark dataset overall and particularly for underrepresented groups.

## 2 Methods

### 2.1 The Fitzpatrick 17k dataset

Images of skin conditions with accompanying diagnostic and skin tone labels are derived from the Fitzpatrick 17k dataset [6]. These data sourced thousands of images from two online dermatology atlases—DermaAmin and Atlas Dermatalogico—and assigned Fitzpatrick skin type (FST) labels to each image using a dynamic consensus process involving 2-5 annotators from Scale AI. The label error rate was estimated at 3.4% relative to board-certified dermatologists. An independent set of annotations were later assigned by a separate team from Centaur Labs [7]. This produced a final dataset comprising 16,577 clinical images. Together, these images represented 114 of the most common dermatology conditions, with a minimum of 53 images per condition.

### 2.2 Selection of skin conditions for retraining models with synthetic images

We selected a subset of seven disease labels from the 114 available in the Fitzpatrick 17k dataset: basal cell carcinoma, folliculitis, neutrophilic dermatoses, prurigo nodularis, squamous cell carcinoma,

Table 1: Sample sizes of seven skin conditions analyzed in this study, by Fitzpatrick skin type

| FST | Basal Cell Carcinoma | Folliculitis | Nematode Infection | Neutrophilic Dermatoses | Prurigo Nodularis | Psoriasis | Squamous Cell Carcinoma | Total |
|---|---|---|---|---|---|---|---|---|
| I | 85 | 30 | 15 | 70 | 7 | 113 | 100 | 420 |
| II | 156 | 97 | 56 | 115 | 28 | 232 | 180 | 864 |
| III | 112 | 99 | 79 | 68 | 39 | 101 | 122 | 620 |
| IV | 76 | 51 | 60 | 51 | 56 | 91 | 71 | 456 |
| V | 24 | 31 | 32 | 31 | 29 | 64 | 40 | 251 |
| VI | 7 | 9 | 12 | 15 | 9 | 21 | 23 | 96 |
| Total | 460 | 317 | 254 | 350 | 168 | 622 | 536 | 2707 |

nematode infection, and psoriasis. These seven conditions were selected as follows: (1) first, we reproduced the analysis by Groh et al. [6] across all 114 conditions in the Fitzpatrick 17k dataset; (2) second, we identified skin conditions that had both the largest sample size at the extremes of FST (I-II or V-VI) and non-zero accuracy when tested on conditions unrepresented in model training. Table 1 lists the sample sizes of each skin condition across FST labels in the dataset.

## 2.3 Synthetic data augmentation using DALL·E 2

We generated photorealistic synthetic images from seed images in the Fitzpatrick 17k dataset using OpenAI's DALL·E 2 model with the following workflow (Figure 1A). For a given skin condition, we randomly sampled eight images from the lightest and darkest Fitzpatrick skin types (16 total images for one condition) to use as seed images. We cropped each seed image with square dimensions centered around the disease pathology. We then utilized DALL·E 2 editor's 'inpainting' function to isolate the primary dermatologic pathology and surrounding skin from background artifacts. The resulting image was used alongside a text prompt to generate synthetic images. Text prompts were produced using the following template: "An image of [skin condition] on the [body part] of a [skin type description] [noun]." All text prompts are listed in Supplementary Table 2. We created eight (on average) synthetic images from each image-text pair, from which we chose four to download and include in our DALL·E 2 training sets. The four images were chosen for photorealism and pathophysiologic consistency. Examples of seed and selected synthetic images are shown in Figure 2. Examples of unselected synthetic images are shown in Supplementary Figure 1.

## 2.4 Model training and evaluation

We trained image classification models to predict skin condition labels among the seven skin conditions using different train and test splits. Following Groh et al. [6], initial models were trained on images of the lightest skin types (FST I-II) and tested on images of darker skin types (FST III-IV and V-VI). The model was also trained on images of the darkest skin types (FST V-VI) and tested on images of lighter skin types (FST I-II and III-IV). These accuracy metrics were used as baselines for comparison against alternate training procedures (Figure 3).

For three skin conditions (squamous cell carcinoma, psoriasis, and neutrophilic dermatoses) we created training sets that included either FST I-II or FST V-VI images from Fitzpatrick 17k supplemented with either 8 seed images from the opposite FST group or both the 8 seed images and 32 DALL·E 2 generated images.

Seed images were not included in any test set. The model architecture and training pipeline, comprising a VGG16 network pre-trained on ImageNet, follows the same structure provided by Groh et al. [6]. Training was performed using Adam optimization and with a weighted random sampler to address class imbalance across skin conditions. Standard image transformations and normalization was applied at the time of training.

## 2.5 Dose-response relationship and spillover effects on classification accuracy

To assess how adding synthetic images of one skin condition affects prediction accuracy across the other skin conditions, we measured model accuracy broadly on all skin conditions in the dataset. For three skin conditions, we also compared models trained on a varying number of synthetic images of the darkest and lightest skin types (2, 8, 16, and 32) to test for a dose-response relationship.

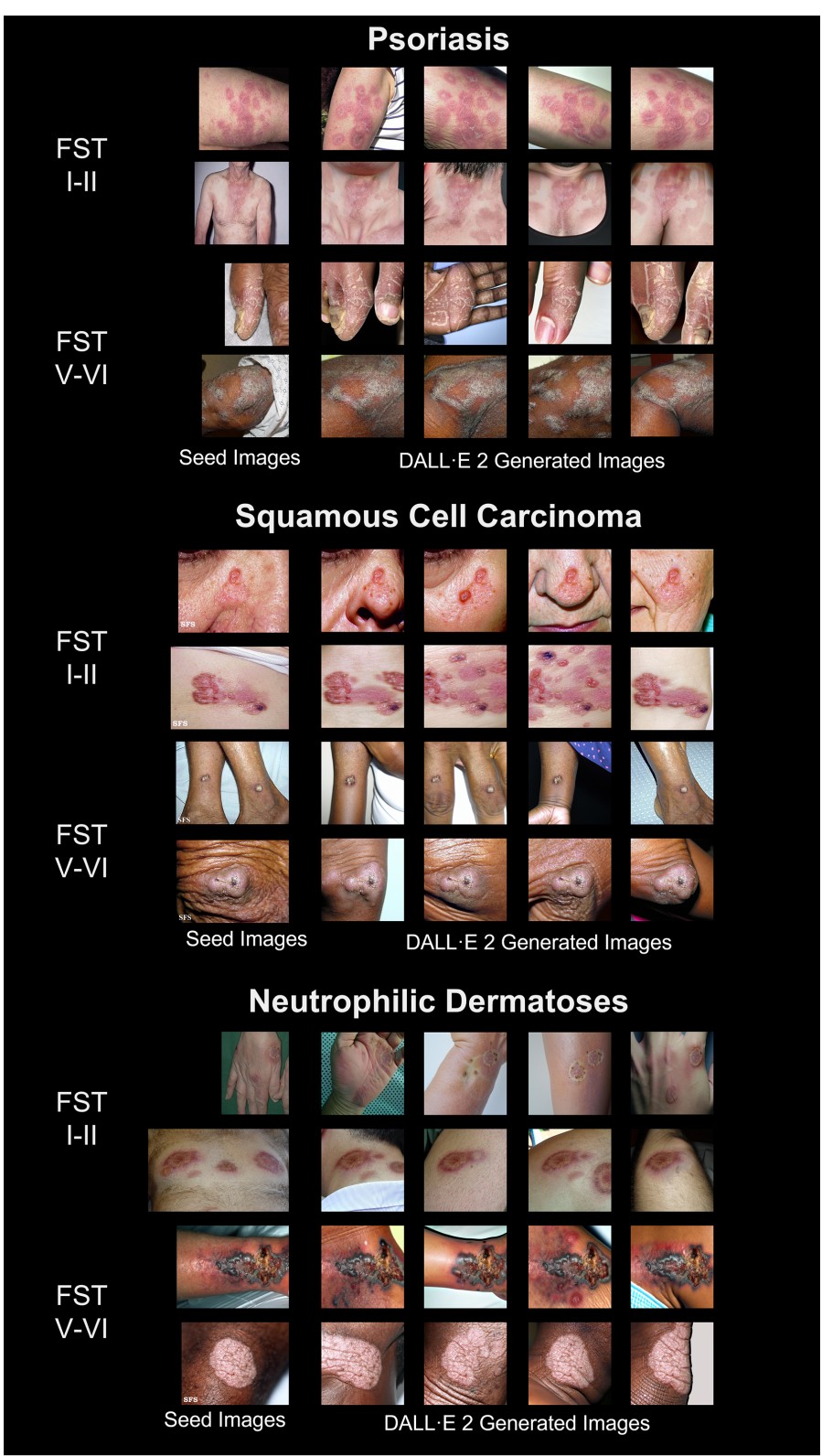

Figure 2: **Examples of DALL·E 2-generated synthetic images** We generated synthetic images for three conditions: psoriasis, squamous cell carcinoma, and neutrophilic dermatoses. For each seed image (left), four synthetic images are shown (right). A full table of text prompts used in these image generations can be found in Supplementary Table 2.

Table 2: Model performance across different real and synthetic training datasets

**Neutrophilic Dermatoses: Classification Accuracy (95% CI), N**

| FST | N | Trained on FST I-II | | | Trained on FST V-VI | | | N |
| | | Fitzpatrick | +Seed | +DALL·E 2 & Seed | Fitzpatrick | +Seed | +DALL·E 2 & Seed | |
| --- | --- | --- | --- | --- | --- | --- | --- | --- |
| I-II | — | — | — | — | 0.175 (0.12-0.23) | 0.310 (0.24-0.38) | 0.610 (0.54-0.68) | 177 |
| III-IV | 119 | 0.311 (0.23-0.39) | 0.395 (0.31-0.48) | 0.411 (0.32-0.50) | 0.311 (0.23-0.39) | 0.378 (0.29-0.47) | 0.571 (0.48-0.66) | 119 |
| V-VI | 37 | 0.243 (0.10-0.38) | 0.594 (0.44-0.75) | 0.675 (0.52-0.83) | — | — | — | — |

**Psoriasis: Classification Accuracy (95% CI), N**

| FST | N | Trained on FST I-II | | | Trained on FST V-VI | | | N |
| | | Fitzpatrick | +Seed | +DALL·E 2 & Seed | Fitzpatrick | +Seed | +DALL·E 2 & Seed | |
| --- | --- | --- | --- | --- | --- | --- | --- | --- |
| I-II | — | — | — | — | 0.249 (0.20-0.29) | 0.359 (0.31-0.41) | 0.504 (0.45-0.55) | 337 |
| III-IV | 192 | 0.495 (0.42-0.57) | 0.557 (0.48-0.63) | 0.573 (0.50-0.64) | 0.255 (0.19-0.31) | 0.370 (0.30-0.44) | 0.463 (0.39-0.53) | 192 |
| V-VI | 77 | 0.753 (0.66-0.85) | 0.857 (0.78-0.94) | 0.857 (0.78-0.94) | — | — | — | x |

**Squamous Cell Carcinoma: Classification Accuracy (95% CI), N**

| FST | N | Trained on FST I-II | | | Trained on FST V-VI | | | N |
| | | Fitzpatrick | +Seed | +DALL·E 2 & Seed | Fitzpatrick | +Seed | +DALL·E 2 & Seed | |
| --- | --- | --- | --- | --- | --- | --- | --- | --- |
| I-II | — | — | — | — | 0.272 (0.22-0.32) | 0.349 (0.29-0.41) | 0.577 (0.52-0.64) | 272 |
| III-IV | 193 | 0.492 (0.42-0.56) | 0.487 (0.42-0.56) | 0.461 (0.39-0.53) | 0.389 (0.32-0.46) | 0.430 (0.36-0.50) | 0.461 (0.39-0.53) | 193 |
| V-VI | 55 | 0.545 (0.41-0.68) | 0.618 (0.49-0.75) | 0.691 (0.57-0.81) | — | — | — | — |

# 3 Results

## 3.1 Dermatological classifiers generalize poorly to underrepresented skin types

We observed that models trained using data from one end of the Fitzpatrick skin-type (FST) scale may exhibit worse performance on skin-types on the opposite end of the FST scale (Table 2). An example of this is seen for neutrophilic dermatoses, where a model trained on images with the lightest FST labels (I-II), exhibited worse performance for the darkest skin types (prediction accuracy for FST V-VI: 24.3%) than for the intermediate skin types (prediction accuracy for FST III-IV: 31.1%). This trend was observed for both models trained on light and tested on dark skin types as well as for models trained on dark and tested on light skin types. In squamous cell carcinoma, for instance, models trained on the darkest skin types (FST V-VI) performed worse on the lightest skin types (prediction accuracy for FST I-II: 27.2%) than on the intermediate skin types (prediction accuracy for FST III-IV: 38.9%). In other instances, the trend was reversed, as in psoriasis, where model performance was better in the darkest FST labels (prediction accuracy for FST V-VI: 75.3%) than in the intermediate labels (prediction accuracy for FST III-IV: 49.5%) when the model was trained on images from the lightest FST groups (FST I-II).

## 3.2 Improved performance for skin disease classification with added synthetic training images

Model performance generally improved when training was supplemented by seed images from unrepresented FST labels and improved further when additionally supplemented by synthetic images generated by DALL·E 2 (Figure 3), although substantial imprecision was observed for conditions with limited test data. The most substantial improvements occurred for images of skin tones least like those on which the model was trained. An example of this can be seen in squamous cell carcinoma for models trained on FST V-VI (bottom right Figure 3). The difference between the performance of the model trained on Fitzpatrick 17k images only and the model trained with Fitzpatrick 17k images plus synthetic and seed images is 7.2% for the FST III-IV group, whereas the performance difference is 30.5% in the FST I-II group (the group furthest from the FST V-VI images).

## 3.3 Secondary analyses showed positive dose-response effect overall and modest spillover effects in non-augmented skin conditions

When we incrementally added 2, 8, 16, and 32 synthetic images to the models, we saw overall increases in model performance in correspondence to the number of synthetic images added, lending evidence toward a dose-response effect (Table 3). In a separate analysis, we generally observed modest changes in classification accuracy for non-augmented skin conditions when synthetic images from one skin condition were added to the training data. The largest difference in performance

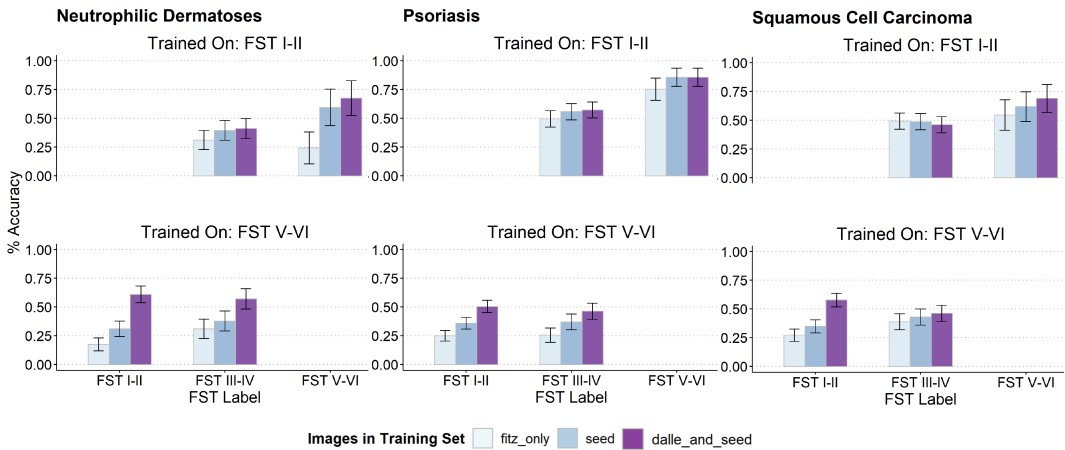

Figure 3: **Model accuracy for different real and synthetic training datasets** Results are shown for three disease labels across models. Each bar represents the performance of a model trained on a subset of Fitzpatrick skin types (e.g. I-II) and tested on the rest of the skin types (e.g. III-IV & V-VI). The color labels represent which images were included in training. "fitz_only" includes only original images from the Fitzpatrick 17K dataset. "seed" includes the original images plus the 8 seed images that were removed from the test set and used in the image generation process. "dalle_and_seed" includes the original images plus synthetic and seed images.

Table 3: Classification accuracy by skin type with successive addition of synthetic training images

| | **Classification accuracy by number of added synthetic training images** | | | | |
|---|---|---|---|---|---|
| **FST** | **+2 images** | **+8 images** | **+16 images** | **+32 images** | **N** |
| | *Neutrophilic Dermatoses* | | | | |
| **III-IV** | 0.37 | 0.34 | 0.39 | 0.45 | 119 |
| **V-VI** | 0.29 | 0.29 | 0.58 | 0.68 | 37 |
| | *Psoriasis* | | | | |
| **III-IV** | 0.49 | 0.56 | 0.56 | 0.51 | 192 |
| **V-VI** | 0.78 | 0.82 | 0.83 | 0.86 | 77 |
| | *Squamous Cell Carcinoma* | | | | |
| **III-IV** | 0.44 | 0.43 | 0.50 | 0.46 | 193 |
| **V-VI** | 0.56 | 0.53 | 0.60 | 0.69 | 55 |

occurred in basal cell carcinoma after light-skinned (FST I-II) synthetic images of squamous cell carcinoma were included in the training set containing only images from FST V-VI. Classification accuracy for basal cell carcinoma in FST I-II decreased from 66.4% to 30.7% (N = 241) and from 68.1% to 50.5% (N = 188) in FST III-IV (Supplementary Table 1).

## 4   Discussion

Development datasets for machine learning models are limited by class imbalances that may limit generalizability for underrepresented groups. We present a proof-of-concept that data augmentation using photorealistic synthetic images of dermatologic pathologies may improve performance across diverse populations in several skin conditions. The results extend upon prior work leveraging deep generative adversarial networks (GANs) [5], style transfer [11], deep blending, or other methods for synthetic data generation.

Performance improvements were observed to follow a dose-response relationship for several skin conditions as synthetic images were added up to a maximum of 32 images (Table 3). Although adding synthetic images primarily affected classification for the skin conditions used to produce them, we observed some instances of performance degradation in non-augmented disease classes (Supplementary Table 1).

Limitations include the limited number of assessed skin conditions, use of a coarse photosensitivity scale to represent skin tone, and requirement for manual involvement in the image generation process. We also cannot rule out the possibility of data leakage from inclusion of Fitzpatrick 17k test data in the DALL·E 2 training data, or from highly similar images in Fitzpatrick 17k. Follow-up work may compare diffusion models head-to-head with previous approaches (e.g., GANs), directly quantify photorealism (e.g., using a Turing test for human clinicians), or investigate the use of synthetic data augmentation to improve robustness to other challenging domains (e.g., lighting conditions or zoom settings).

With the DALL·E 2 application programming interface (API) now available alongside the already open-source Stable Diffusion model, developing nearly or fully automated pipelines for prompting, generating, and selecting synthetic images at scale may soon be feasible. The creation of large, expert-vetted repositories of synthetic data could improve access to diverse training data while mitigating privacy concerns. In addition, investigators could study the effect of adding synthetic images in numbers comparable to or greater than the training data. Of note, a recent study of glaucoma detection showed that models trained exclusively on synthetic images generated by GANs resulted in similar performance on external test sets as models trained exclusively on real images [11].

While collection of diverse real-world data remains the most important and rate-limiting step for improving skin classification models, we believe that the concomitant use of synthetic data, along with traditional methods of re-weighting and upsampling, may act as a force-multiplier to continually improve classification models for skin pathology.

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
