# OpenReview forum: "Improving dermatology classifiers across populations using images generated by large diffusion models"
_NeurIPS.cc/2022/Workshop/SyntheticData4ML — Neurips 2022 SyntheticData4ML_

### Official Review · Reviewer_soWh · 2022-10-11
**The authors present an interesting application of DALLE-2, for dermatology-related images to improve the performance of downstream tasks for underrepresented groups.  I believe that the work could be improved by comparing the performance to other related models.**

**Rating:** 7
**Confidence:** 4

**Review:**


Pros:
1) The authors present an interesting application of DALLE-2 that addresses a gap in the lack of data for underrepresented groups.
2) The authors show proof of concept and improvement in results across multiple skin diseases and conduct an ablation study to demonstrate further the improvements achieved by their proposed approach.

Cons:
1) Despite the promise of the author's work, I believe this work could also be improved by comparing it with other state-of-the-art generative models with conditional variants, such as GAN.
2) Furthermore, the data augmentation framework could also be improved by exploring the addition of more synthetic images.

---

### Official Review · Reviewer_zReq · 2022-10-17
**The authors use dall-e to generate images of skins with various disease symptoms. They further obtain improved performance on skin disease classification by better representing under-represented skin tones.**

**Rating:** 6
**Confidence:** 3

**Review:**

Quality
I think that the writing and the evaluation should be improved.

Please see my suggestions for writing below:
Abstract
While more intentional
data collection and annotation is the best way to increase representation, new
computational approaches for generating training data -> While intentional
data collection and annotation is the best way to increase representation, new
computational approaches for generating training data

including DALL·E 2-generated synthetic images improves classification accuracy -> including DALL·E 2-generated synthetic images in training data improves classification accuracy

Introduction
the performance for dermatological classifiers on a -> the performance of dermatological classifiers on a

Evaluation:
Why do you report the results for only 3 skin conditions, while you train models to predict among 7 conditions?

Table 3 can be made a plot.

No comparison with the alternative deep generative models is made.

Originality
As far as I could understand, the work is novel in the sense that no-one used DALL-E to generate synthetic data for skin disease classification before.

Significance
The data for skin disease classification seems to be limited. Therefore, synthetically generating realistic data can help applying ML techniques. It is particularly important for skin types under-represented in training data.

---

### Official Review · Reviewer_KDU7 · 2022-10-18
**An interesting investigation that motivates a direction of future research**

**Rating:** 7
**Confidence:** 3

**Review:**

I enjoyed reading this submission. This paper presents a clear and easy-to-follow investigation of large-scale diffusion models’ potential to generate synthetic data for dataset enrichment and model performance improvement by demonstrating performance improvements in the task of skin disease classification. I have recently been excited by the recent developments of diffusion model technology, such as DALL E 2, and this paper inspires the use of such technology as a valid means of promoting model performance and fairness.

There are some research directions following this paper that I wish to see toward a more robust case of validating diffusion models for synthetic data generation, but I understand that may not be reasonable given the page limit. However, this submission produces a useful application of large diffusion models to achieve model accuracy improvements in the dermatological setting and sets up further research work to apply and better understand diffusion models in clinical and other contexts for synthetic data generation.

## Summary

This paper aims to explore the capabilities of emerging developments in large-scale diffusion models to improve model performance that may be limited by factors such as dataset imbalances or class scarcities. This research paper demonstrates the potential of synthetic data generated by large diffusion models to enrich dataset quality and improve model performance in the context of dermatological classification. The authors utilize DALL-E 2, a text-to-image diffusion model, for the directed generation of synthetic, photorealistic images of different skin diseases across varied skin colors. In their synthetic data generation process, they randomly sampled images of selected skin diseases from a publicly-available dataset to use as seed images. These seed images were complemented with a carefully-structured text prompt as input to DALL-E 2, yielding several synthetic variations as output. Their experiments show that the accuracy of skin disease classification models generally improves overall, and especially for underrepresented groups, when trained on a skin disease dataset supplemented by the artificially generated data from DALL-E 2, compared to training on the original dataset alone.

This paper provides an interesting study on the potential of recent advancements in diffusion technology to enable dataset enrichment and to help address challenges such as data scarcity or class under-representation. Their application of purposely generated images from DALL-E 2 to improve skin disease classification accuracy not only demonstrates the potential for synthetic data to promote model fairness, but shows the capability of large diffusion models to be a valid means of generating data to help achieve dataset quality, quantity, and diversity, especially in contexts where data collection may be more scarce or difficult to collect in nature.

## Originality

The paper discusses the concerns of limited representation and impacted performance of clinical classification models in underrepresented groups and cites the existing approaches to address such concerns: manual collection of developmental datasets with greater diversity/representation, use of generative adversarial networks technology, and other deep learning methods for synthetic data generation. However, this paper takes a unique approach to synthetic data generation by leveraging a diffusion model, namely DALL-E 2, to enrich training dataset quality in the dermatological context, demonstrating classification accuracy improvement and claiming performance improvements that had not yet been achieved by previously mentioned deep learning methods. While it is undescribed how performance improvement per synthetic data generation via DALL-E 2 compares to other deep learning data generation/augmentation methods, the paper provides a clear, distinctive process of generating synthetic data via diffusion models and demonstrates the validity of artificial data generation via diffusion technology for dataset enhancement.

## Quality

The methods, techniques, and processes described in this study are appropriate in the interest of exploring synthetic data generation via large diffusion models. The process of data generation is structured and designed to be consistent across image generation samples. The authors provide details about images/prompts that were included/not included (see supplementary info) in the training data. However, it is unclear if there is a specific criterion for filtering generated images for inclusion in the training set. The described training and evaluation of model performance on the selected datasets (original data w/out, original data w/ seed, original data + seed + generated images) is directed and limits the prevalence of confounding variables to permit an explicit consideration of the influence of generated data on model performance. The authors discuss the limitations of their methods (data leakage/inclusion in DALL E 2, significant manual involvement in the data generation process, etc) and discuss their evaluations in the broad context of dataset development (synthetic data can enhance model performance but won’t replace real-world data).

## Clarity

This submission is clearly written and easy to follow. The relevant background and context of the paper are well described and the authors’ vision to generate synthetic data using diffusion models for dermatological model classification is well communicated. The process of data generation, including the filtering of generated images removed from the training set (described in the supplementary info), is well-described, transparent, and easy to reproduce. While it is apparent that images have been filtered out from inclusion in the dataset, it is unclear if/how the generated images were assessed for validity/quality. The separate datasets that were created for model training and evaluation are distinguished and explicit. The details on model training and evaluation are defined and succinct as the model is built on previous work done on the same task (cited from M. Groh, citation [5] during the time of writing). The results show a clear general improvement of the model performance in both overall and particular group classification accuracy.

## Significance and Follow-Up

The results from this submission provide an insightful investigation into the potential for diffusion models like DALL-E 2 to be a valid means of generating synthetic data to improve model performance in the clinical setting, as well as in general contexts. Researchers are likely to be inspired to use DALL-E 2 and other diffusion models in similar ways to promote dataset enrichment and improve model performance. This submission does not attempt to investigate other questions beyond the potential for improved model performance via the use of generated data from diffusion models in ML, such as: How do synthetic images influence particular attributes of model training to impact performance/accuracy, How can generated images be used to better understand and address the limitations in models, How can the generation/screening of the data generation be better streamlined or more carefully formatted for larger scale experiments, How to deliberately craft effective queries, etc. These are some of the research questions inspired by this submission and may prompt follow-up exploration for a more thorough understanding of how to use large diffusion models for model enhancement. Nevertheless, this paper does a good job of exciting the potential for new diffusion models like DALL-E 2 to enhance dataset quality/model performance by demonstrating model classification accuracy improvements in the dermatological setting.

---

### Meta-Review · Area_Chair_RWvy · 2022-10-18

**Recommendation:** Accept